# Costs, effectiveness, and safety associated with Chimeric Antigen Receptor (CAR) T-cell therapy: Results from a comprehensive cancer center

**Sérgio Chacim**[1,2☯], **Teresa Monjardino**[3☯]*, **José Luís Cunha**[4,5], **Pedro Medeiros**[4,5,6], **Patrícia Redondo**[4,5], **Maria José Bento**[3,7,8], **José Mário Mariz**[1]

1 Department of Onco-hematology, Portuguese Oncology Institute of Porto (IPO-Porto), Porto, Portugal, 2 Cancer Biology and Epigenetics Group, Portuguese Oncology Institute of Porto Research Center (CI-IPOP) / RISE@CI-IPOP (Health Research Network) / Porto Comprehensive Cancer Center (Porto.CCC), Porto, Portugal, 3 Cancer Epidemiology Group, Portuguese Oncology Institute of Porto Research Center (CI-IPOP) / RISE@CI-IPOP (Health Research Network) / Porto Comprehensive Cancer Center (Porto.CCC), Porto, Portugal, 4 Outcomes Research Lab, Portuguese Oncology Institute of Porto (IPO-Porto), Porto, Portugal, 5 Management, Outcomes Research, and Economics in Healthcare Group, Portuguese Oncology Institute of Porto Research Center (CI-IPOP) / RISE@CI-IPOP (Health Research Network) / Porto Comprehensive Cancer Center (Porto.CCC), Porto, Portugal, 6 Medicine and Oncological Medicine Departments Management, Portuguese Oncology Institute of Porto (IPO-Porto), Porto, Portugal, 7 Department of Epidemiology, Portuguese Oncology Institute of Porto (IPO-Porto), Porto, Portugal, 8 Department of Population Studies, ICBAS-School of Medicine and Biomedical Sciences, University of Porto (ICBAS-UP), Porto, Portugal

☯ These authors contributed equally to this work.

* teresa.monjardino@ipoporto.min-saude.pt

## Abstract

Real world effectiveness, toxicity and costs analyses from chimeric antigen receptor (CAR)-T cell therapy are of utmost relevance to determine whether and how to offer patients highly personalized immunotherapy. In this study, we aimed at describing CAR T-cells effectiveness, safety and costs in a Portuguese Comprehensive Cancer Center. We performed a retrospective descriptive study of adult patients with relapsed/refractory diffuse large B-cell lymphoma (DLBCL), primary mediastinal B-cell lymphoma and transformed follicular lymphoma referred to CAR T-cell therapy, between May 2019 and February 2021. Rates of treatment response, toxicity and survival (Kaplan-Meier method) were analyzed by intention-to-treat. Direct medical costs stratified by inpatient-care, outpatient-care, and diagnostic-therapeutic procedures (DTP) were derived based on resources used and their respective unit costs. In twenty patients (median age 49.5y; 55%male; 70%DLBCL; 50% with primary refractory disease), best overall and complete response rates were 65.0% and 45.0%, respectively. Median overall (OS) and progression-free survivals were 9.2 and 7.3 months; 12-month OS rate was 42.6% (95%CI:23.2–78.3). Grade≥3 cytokine release syndrome and neurotoxicity occurred in 5.6% and 11.1% of patients, respectively. CAR T-cell therapy expenditure, including adverse events costs, was 7 176 196€, or 286 238€ when excluding drug cost. Median cost for treated patient was 355 165€ with CAR T-cell drug cost accounting for 97.0% of the overall expense. Excluding CAR T-cell acquisition cost,

**Data Availability Statement:** Due to imposition by the Data Protection Officer from Portuguese Oncology Institute of Porto (IPO-Porto), data from

this study is only available upon request due to legal and ethical restrictions on sharing data publicly. IPO-Porto Data Protection Officer provided the following explanation: "This study is based on a very small sample (n=20), so we understand that the particularity of this study, given the current state of the art, may lead to an increased risk of violating the privacy of our patients through a possible identification of the data subject. However, this concern is the result of the ethical and legal obligations to which our institution is bound in light of the current guidelines and recommendations and legislation". Data requests may be sent to the Data Protection Officer (epd@ipoporto.min-saude.pt) and to the Department of Onco-hematology at IPO-Porto (s. oncohematologia@ipoporto.min-saude.pt).

**Funding:** The author(s) received no specific funding for this work.

**Competing interests:** The authors have declared that no competing interests exist.

inpatient-care and DTP accounted for 57% and 38% of total cost/patient, respectively. Our findings highlight the heavy economic burden of CAR T-cell therapy driven by drug acquisition costs.

## Introduction

Chimeric antigen receptor (CAR)-T cells represent a new class of cancer immunotherapies that genetically engineer patient T-cells to target their disease [1, 2]. Results from CAR T-cells clinical trials and real-life evidence have shown high rates of durable remissions responses and meaningful overall survival benefits in relapsed/refractory large B-cell lymphomas [3–10]. However, these therapies are priced amongst the most expensive cancer therapies to date with list prices of approximately $373 000 US dollars in US and 320 000€ in Europe [11]. In addition to direct costs of acquisition and infusion of CAR T-cells, lymphodepletion, outpatient visits and exams, there are also costs attributable to hospitalization, intensive care unit (ICU) admissions, laboratory activity, imaging studies, specialized and multidisciplinary teams work and management of potentially life-threatening adverse effects [11–18]. Nevertheless, substitution effects may reduce the financial impact or CAR T-cell therapy, avoiding current futile treatments, potentially reducing autologous and allogenic stem cell transplantation for relapsed patients. Formal assessment of these aspects will improve knowledge about this technology, effective economic evaluation and understanding its real cost [11, 19, 20].

To date, scarce research has quantified real costs attributable to CAR T-cell therapy, which is crucial in drug reimbursement discussion between health players. Country-specific economic evaluations are necessary to determine whether and how to offer patients these highly personalized forms of immunotherapy [11, 20–24].

Therefore, this study aimed to describe effectiveness and safety and estimate the costs associated with CAR T-cell therapy in a real-life clinical setting, the Portuguese Comprehensive Cancer Center of Portuguese Oncology Institute of Porto (IPO-Porto).

## Materials and methods

### Patients

We performed a real-world descriptive study with retrospective data collection for all adult patients with relapsed/refractory aggressive diffuse large B-cell lymphoma (DLBCL), primary mediastinal B-cell lymphoma (PMBCL) and transformed follicular lymphoma (TFL) who underwent leukapheresis with the intent to receive CAR T-cell therapy—Axicabtagene Ciloleucel (axi-cel) and Tisagenlecleucel (tisa-cel)—between May 2019 and February 2021, with follow-up until March 2021, in IPO-Porto.

Patients with histologically confirmed DLBCL, PMBCL or TFL were eligible for CAR T-cell therapy according to predefined inclusion/exclusion criteria presented in Table 1.

### Data collection

Demographic, clinical, treatment, effectiveness, toxicity, healthcare resources utilization and costs data were abstracted from clinical and administrative records. Bridging therapy was defined as any anti-cancer therapy administered after leukapheresis but prior to CAR T-cell infusion. Tumor response assessment were performed per Lugano 2014 classification criteria [25]. Cytokine release syndrome (CRS) and immune effector cell-associated neurotoxicity

**Table 1. Baseline patient characteristics [a].**

| Characteristic | Total (n = 20) |
|---|---|
| Median [min–max] age (years) | 49.5 [23.3–66.1] |
| Patients ≥ 60 years | 5 (25.0) |
| Sex (male) | 11 (55.0) |
| ECOG PS at leukapheresis | |
| 0 | 12 (60.0) |
| 1 | 8 (40.0) |
| IPI at lymphodepletion [b] | |
| 1 | 8 (40.0) |
| 2 | 7 (35.0) |
| 3 | 4 (20.0) |
| 4 | 1 (5.0) |
| Diagnosis | |
| DLBCL | 14 (70.0) |
| PMBCL | 3 (15.0) |
| TFL | 3 (15.0) |
| Disease stage at diagnosis | |
| I | 2 (10.0) |
| II | 6 (30.0) |
| III | 3 (15.0) |
| IV | 9 (45.0) |
| Double/triple-hit molecular subtype | 3 (16.7) |
| Missing | 2 |
| Cell of origin in DLBCL patients (n = 14) | |
| GCB | 8 (80.0) |
| Non-GCB | 2 (20.0) |
| Missing | 4 |
| Tumor bulk, cm | |
| <5 | 12 (60.0) |
| 5–10 | 6 (30.0) |
| >10 | 2 (10.0) |
| LDH > 1.5X ULN at CAR T-cell referral | 7 (36.8) |
| Missing | 1 |
| LDH > 1.5X ULN at CAR T-cell infusion (n = 18) | 4 (22.2) |
| Median [min-max] no. of lymphocytes at CAR T-cell infusion ($*10^9$ / mcL) (n = 18) | 0.04 [0.01–0.53] |
| Missing | 2 |
| History of primary refractory disease | 10 (50.0) |
| Prior ASCT | 3 (15.0) |
| Previous lines of therapy | |
| 2 | 10 (50.0) |
| 3 | 9 (45.0) |
| 4 | 1 (5.0) |
| Median [min-max] time from previous line of therapy last cycle to CAR T-cell referral (days) | 49 [0–1324] |
| Median [min-max] time from CAR T-cell referral to CAR-T cell infusion (days) (n = 18) | 65 [43–121] |
| Median [min-max] time from leukapheresis to CAR T-cell infusion (days) (n = 18) | 41 [32–74] |
| Bridging | 12 (60.0) |
| Bridging therapy (n = 12) | |
| PEPC | 7 (58.3) |

*(Continued)*

**Table 1.** (Continued)

| Characteristic | Total (n = 20) |
|---|---|
| Radiotherapy | 2 (16.7) |
| GEMOX | 1 (8.3) |
| HCVAD | 1 (8.3) |
| R-Benda | 1 (8.3) |
| CAR T-cell therapy | |
| Axicabtagene ciloleucel | 13 (65.0) |
| Tisagenlecleucel | 7 (35.0) |

ECOG PS, Eastern Cooperative Oncology Group performance status; IPI, international prognostic index score; DLBCL, diffuse large B-cell lymphoma; PMBCL, primary mediastinal B-cell lymphoma; TFL, transformed follicular lymphoma; GCB, germinal center B cell; LDH, lactate dehydrogenase; ULN, upper limit of normal; CAR, chimeric antigen receptor; ASCT, autologous stem cell transplantation; PEPC, cyclophosphamide, etoposide, prednisolone and procarbazine; GEMOX, gemcitabine/oxaliplatin; HCVAD, hyper-cyclophosphamide, vincristine, doxorubicin, dexamethasone; R-Benda, rituximab-bendamustine.

Data are presented as No. (%) unless indicated otherwise.

[a] Patients with histologically confirmed DLBCL, PMBCL or TFL were eligible for CAR T-cell therapy if they met the predefined inclusion criteria: age between 18 and 65 years; Eastern Cooperative Oncology Group (ECOG) performance status (PS) at leukapheresis of 0 or 1; relapsed or refractory disease after two lines of chemotherapy or within 12 months of autologous stem-cell transplantation; absolute neutrophil count of at least 1000/μL; absolute lymphocyte count of at least 100/μL; platelet count of at least 75 000/μL; adequate renal and hepatic function; forced expiratory volume $\geq$ 50% without pericardial effusion and oxygen saturation $\geq$ 92% in room air. Exclusion criteria were as follows: history of other malignancy (other than non-melanoma skin carcinoma or carcinoma "in situ" of the cervix, bladder or breast in the last 3 years); Richter's transformation from chronic lymphocytic leukemia; previous allogeneic transplantation; active infection; human immunodeficiency virus (HIV), hepatitis B (HBsAg positive) or hepatitis C (anti-HCV positive) infection; current or history of central nervous system (CNS) involvement; history of seizure crisis, ischemic or hemorrhagic cerebrovascular disease, dementia, cerebellar disease, autoimmune disease with CNS involvement; heart involvement; need for urgent secondary treatment due to large tumor mass and primary immunodeficiency.

[b] Lymphodepletion chemotherapy for Axicabtagene ciloleucel

- intravenous (IV) ondansetron (8 mg) every 8 hours, at days -5, -4 and -3
- IV saline solution (2000 ml) per day, at days -5, -4 and -3
- IV cyclophosphamide (500 mg/m$^2$) infused over 60 minutes, at days -5, -4 and -3
- IV fludarabine (30 mg/m$^2$, maximum dose of 50 mg) infused over 30 minutes, at days -5, -4 and -3
- Lymphodepletion chemotherapy for Tisagenlecleucel:
- IV ondansetron (8 mg) every 8 hours, at days -5, -4 and -3
- IV saline solution (2000 ml) per day, at days -5, -4 and -3
- IV cyclophosphamide (250 mg/m$^2$) infused over 60 minutes, at days -5, -4 and -3
- IV fludarabine (25 mg/m$^2$, maximum dose of 50 mg) infused over 30 minutes, at days -5, -4 and -3

syndrome (ICANS) were graded by the American Society for Blood and Marrow Transplantation (ASBMT) Consensus Grading for Cytokine Release Syndrome and Neurologic Toxicity Associated with Immune Effector Cells [26, 27].

Costs data associated with CAR T-cell treatment, resources used in inpatient care, including drugs consumed during hospital stay, outpatient care, including medical and non-medical appointments, and diagnostic and therapeutic procedures were collected to estimate the direct medical costs.

Referral date to CAR T-cell therapy was considered the starting point (baseline) for data collection and medical resource consumption accountability.

## Cost estimation

Direct medical costs attributable to CAR T-cell therapy, total and stratified by resources use categories—CAR T-cells pharmaceutical drug, outpatient care (including hospital medical and non-medical appointments), inpatient care (in Hematology Department or in ICU) and diagnostic and therapeutic procedures—were derived based on healthcare resources used by the application of unit costs.

The micro-costing technique, where all medical direct costs are individually accounted for each patient at each stage of treatment, from the referral date to the date of the last observation in the follow-up period, was used. Analysis was stratified by stages in CAR T-cell treatment course as follows:

- Stage 1: From referral date to CAR T-cell therapy until the day before infusion

- Stage 2: From infusion day until day 30 after infusion

- Stage 3: Between 31 and 60 days after infusion

- Stage 4: Between 61 and 150 days after infusion

In each stage of the treatment course, the activity of patients that completed treatment stage or succumbed to the diseases during the given stage was analyzed. Patients that were still completing the given stage were excluded to avoid skewing the data. Unit costs for appointments (medical and non-medical), exams and procedures were obtained from government official prices [28]. Inpatient care costs were obtained from IPO-Porto analytical accounting for the year of 2019. Total costs of hematology hospitalizations were divided by the length of stay of patients to obtain the cost per day. Drug prices were obtained as the weighted average of the real cost for each pharmaceutical used in IPO-Porto, which follows the approved prices by the Portuguese government agency regulating medicines (INFARMED, IP).

Only medical direct costs from the healthcare provider perspective were analyzed meaning that discount rate and initial early access programs for CAR T-cell technology, which accommodate lower prices or even free of charge, were not considered.

## Statistical analysis

Effectiveness outcomes were reported in an intent-to-treat (ITT) analysis on all patients who underwent leukapheresis with the intention to receive CAR T-cell therapy. Overall survival (OS) was defined as from the date of the referral to CAR T-cell therapy to death from any cause, censoring for patients alive at last contact. Progression-free survival (PFS) was defined as from CAR T-cell therapy referral date to the earlier of progression or death, censoring for patients alive and progression free at last contact. Kaplan-Meier method was used to estimate PFS and OS rates for all patients who underwent leukapheresis. Rates of toxicity and safety data were calculated in patients who received CAR T-cell therapy. To determine total cost per patient, costs accrued for each variable were summed. Total cost was calculated as the sum of each treated patient costs. Statistical analysis was performed using the software R (version 3.6.2).

## Ethics approval

The study was performed in accordance with the Declaration of Helsinki. The study was approved by the Ethics Committee of IPO-Porto (ref. CES: 188/020). Patient informed consent was not required by the ethics committee due to the retrospective observational nature of this study and because the data contained no unique personal identifiers.

## Results

As of March 2021, 20 patients completed leukapheresis with intention to manufacture and receive CAR T-cell infusion. Patient characteristics are listed in Table 1. Median age was 49.5 years (range, 23.3–66.1 years). DLBCL was the most common lymphoma (70%), followed by PMBCL (15%). From 20 patients who underwent leukapheresis, 13 (65.0%) were selected for treatment with axi-cel and 7 (35.0%) with tisa-cel. Two patients (10.0%) were not treated with CAR T-cells due to disease progression (one from axi-cel treatment group and another from tisa-cel group). The median time from leukapheresis to CAR T-cell infusion was 41 days (range, 32–74 days).

Initial response assessment was performed on 16 of 18 infused patients (for 13 patients at month 1 and for 3 patients at month 3). Two patients died before any response assessment, one died because of toxicity and the other due to disease progression (S1 Fig). Five in six patients with partial response (PR) at month 1 were evaluated for subsequent response at month 3: 2 patients achieved CR, 1 remained with PR and in 2 patients the disease progressed. Median [min-max] follow-up time from CAR T-cell infusion was 4.0 [0.2, 13.9] months. Table 2 reviews effectiveness outcomes by ITT. The best overall response rate (ORR) was 65.0% (95%CI: 40.8 to 84.6) and 9 patients (45.0%; 95%CI: 23.0 to 68.5) achieved CR.

Median OS for the 20 patients who underwent leukapheresis was 9.2 months (95%CI: 6.3 to not reached (NR)), from the date of referral to CAR T-cell therapy, and 12-month OS rate was 42.6% (95%CI: 23.2 to 78.3) (Fig 1). Median PFS was 7.3 months (95%CI: 5.3 to NR) and 12-month PFS rate was 37.4% (95%CI: 18.8 to 74.5) (Fig 2).

Any grade CRS occurred in 72.2% of patients, including $\geq$ grade 3 in 1 patient (5.6%). Immune effector cell-associated neurotoxicity syndrome (ICANS) of any grade and grade $\geq$ 3 occurred in 27.8% and 11.1% of patients, respectively. One death occurred because of ICANS. ICU admission occurred for 16.7% of patients and median ICU stay was 8 days (range, 6–10 days) (Table 3).

Healthcare resources utilization by categories of resources is presented in Table 4. We observed 11 pre-infusion hospitalizations, with a median of 2.0 days of length of stay, in 7 of the 20 patients eligible to receive CAR T-cell infusion in IPO-Porto. Median length of stay of the infusion hospitalizations was 17 days. In 18 patients infused with CAR T-cells, 3 were hospitalized 4 times (median length of stay of 17.5 days).

**Table 2. Effectiveness outcomes.**

| Characteristic | Total (n = 20) |
|---|---|
| Overall survival | |
| Alive with no evidence of disease | 8 (40.0) |
| Alive with evidence of disease | 3 (15.0) |
| Fatal events | 9 (45.0) |
| Disease progression | 7 (35.0) |
| ICANS | 1 (5.0) |
| Hemophagocytic lymphohistiocytosis | 1 (5.0) |
| Best Overall response rate [a] | 13 [65.0 (95%CI: 40.8–84.6)] |
| Complete response | 9 [45.0 (95%CI: 23.0–68.5)] |
| Partial response | 4 [20.0 (95%CI: 5.7–43.7)] |
| Disease progression | 3 [15.0 (95%CI: 3.2–37.9)] |

ICANS, immune effector cell-associated neurotoxicity syndrome; 95%CI: 95% confidence interval.

[a] Two patients died before infusion and 2 patients died before response assessment.

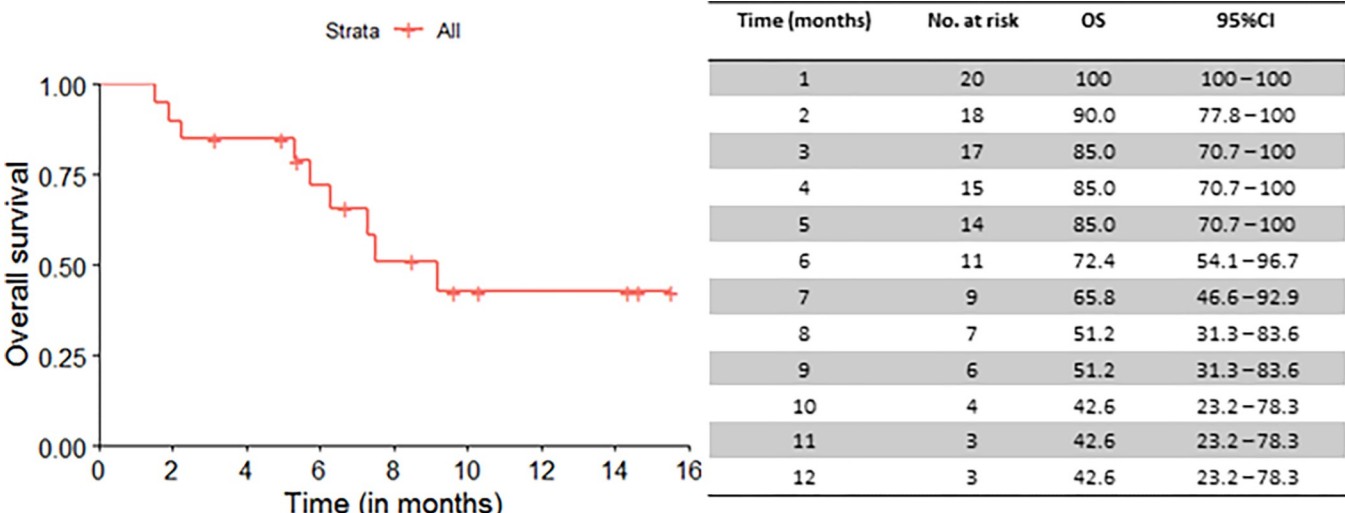

**Fig 1. Overall survival curve for all patients (n = 20).** Median OS (months): 9.2 (95%CI: 6.3 to not reached).

Costs of CAR T-cell treatment, total, by categories of resources use and by stages of treatment, are presented in Table 5. For the 20 patients eligible to CAR T-cell treatment, total medical cost of CAR T-cell therapy, including expenditure with adverse events, was 7 176 196€. The overall IPO-Porto expenditure with these patients substantially decreased to 286 238€ when CAR T-cell acquisition cost was not included in the estimation. Substantially higher healthcare costs for the institution were observed in the earlier stages of treatment course, from referral until day 30 after infusion, than later, between 31 and 150 days after infusion. As the number of patients decreased from 20 in stage one to 13 in stage 4, proportions of the sub-total costs (excluding CAR T-cell drug cost) declined from 36.9% in stage 1, 42.3% in stage 2, 15.2% in stage 3 and 5.6% in stage 4.

Costs of CAR T-cell treatment, total and by categories of healthcare resource, are presented in Table 6. Our results showed a median cost for treated patient of 355 165€, which about

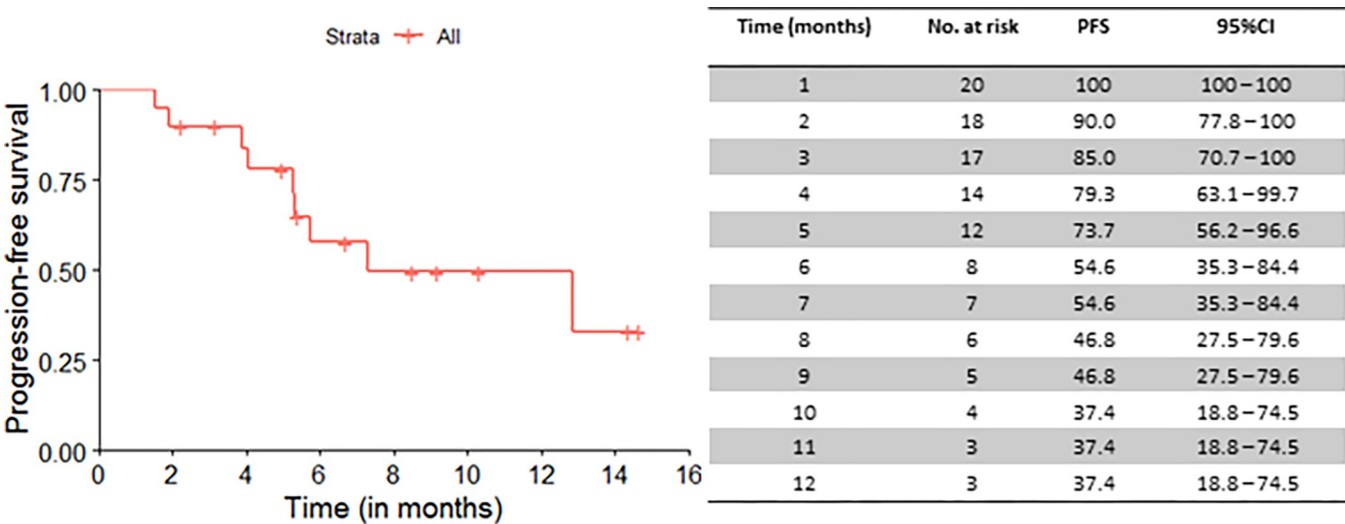

**Fig 2. Progression-free survival curve for all patients (n = 20).** Median PFS (months): 7.3 (95%CI: 5.3 to not reached).

**Table 3. Safety: Toxicity events, grade and treatment.**

| Characteristic | Total (n = 18) |
|---|---|
| CRS (any grade) | 13 (72.2) |
| CRS grade ≥ 3 | 1 (5.6) |
| Median [min-max] time to CRS onset (days) (n = 13) | 2 [0–5] |
| Median [min-max] duration of CRS (days) (n = 13) | 5 [1–10] |
| ICANS (any grade) | 5 (27.8) |
| ICANS grade ≥ 3 | 2 (11.1) |
| Median [min-max] time to ICANS onset (days) (n = 5) | 5 [2–6] |
| Median [min-max] duration of ICANS (days) (n = 5) | 5 [3–9] |
| Concomitant CRS/ICANS | 5 (27.8) |
| Another AE [a] grade ≥ 3 | 1 (5.6) |
| ICU admission | 3 (16.7) |
| Median [min-max] ICU stay (days) (n = 3) | 8 [6–10] |
| Corticosteroid use | 11 (61.1) |
| Low dose [b] (n = 11) | |
| CRS alone | 10 (90.9) |
| ICANS alone | 0 |
| Concomitant CRS/ICANS | 1 (9.1) |
| High dose [c] (n = 4) | |
| CRS alone | 0 |
| ICANS alone | 4 (100) |
| Concomitant CRS/ICANS | 0 |
| Tocilizumab use | 6 (40.0) |
| Missing | 3 |
| Tocilizumab ≥ 2 doses use | 2 (13.3) |
| Siltuximab/anakinra use | 1 (5.6) |

CRS, cytokine release syndrome; ICANS, immune effector cell-associated neurotoxicity syndrome; AE, adverse event; ICU, intensive care unit.

Data are presented as No. (%) unless indicated otherwise.

[a] Other adverse event included catheter-related infection by Staphylococcus epidermidis.

[b] Low dose corticosteroid use defined as dexamethasone up to 10 mg every 6 hours or equivalent dose.

[c] High dose corticosteroid use defined as 1000 mg of intravenous methylprednisolone as a single day dose or 1 mg/kg every 12 hours.

97.0% of the total medical cost correspond to CAR T-cells drug costs. After excluding CAR T-cell acquisition cost (344 498€), the main driver of higher costs was inpatient care, including hematology and ICU hospitalizations, which account for 57.5% of these costs. Diagnostic and therapeutic procedures cost account for 37.5% of the total, with blood tests and exams and cell therapy procedures representing 15.6% and 9.9%, respectively, of sub-total medical cost.

## Discussion

This real-world cost analysis covers a growing concern on health systems, the burden of expenses related to CAR T-cell therapies. CAR T-cells appear to provide significant clinical benefit despite their high cost, thus making economic evaluations highly relevant to support policymakers in reimbursement decisions [20, 22, 23, 29, 30]. The relevance of this study should be also viewed in the light of continuously evolving indications for this therapy [11, 17, 31–34]. CAR T-cells have already changed the treatment paradigm for relapsed/refractory

**Table 4. Healthcare resources utilization.**

| Activity | Total (n = 20) |
|---|---|
| Median [min-max] no. of medical and non-medical appointments | 19 [12–39] |
| Inpatient care | |
| No. of pre-infusion hospitalizations (n = 7) | 11 |
| Median [min-max] length of stay (days) of pre-infusion hospitalizations (n = 7) | 2.0 [1.0–7.0] |
| No. of infusion hospitalizations (n = 18) | 18 |
| Median [min-max] length of stay (days) of infusion hospitalizations (n = 18) | 17.0 [10.0–35.0] |
| No. of pos-infusion hospitalizations (n = 3) | 4 |
| Median [min-max] length of stay (days) of pos-infusion hospitalizations (n = 3) | 17.5 [2.0–32.0] |
| Median [min-max] no. diagnostic and therapeutic procedures | |
| Imaging Exams (CT scan, PET-CT, MRI, bone scintigraphy) | 2 [0–4] |
| Cell therapy procedures | 2 [1–3] |
| Radiotherapy sessions | 1 [0–2] |
| Others (blood tests and exams) | 32 [14–158] |

CT, computed tomography; PET-CT, Positron Emission Tomography—Computed Tomography; MRI, Magnetic Resonance Imaging.

large B-cell lymphomas [35]. Our efficacy results are comparable to those obtained in comprehensive meta-analyses summarizing axi-cel and tisa-cel efficacy [9, 10]. Frequency of adverse effects described for our cohort are very similar to those reported for ZUMA-1 and JULIET trials, as well as for other real-life cohorts [3, 5, 7, 8, 36, 37]. High-level CRS and ICANS (grade ≥ 3) was observed in 5.6% (n = 1) and 11.1% (n = 2) of the patient, respectively, findings that are within the range of toxicity rates reported by recent meta-analyses [10]. The safety data highlights the manageable tolerability of this therapy as only 3 patients (16.7%) required an ICU admission with 8 days of median stay. The ICU admission rate at IPO-Porto was lower than a rate of 32% estimated in a hospital-wide retrospective study conducted in France

**Table 5. Costing by stage.**

| Activity | Stage 1 | Stage 2 | Stage 3 | Stage 4 | Total |
|---|---|---|---|---|---|
| | n = 20 | n = 18 | n = 14 | n = 13 | |
| Outpatient care | 7 228€ | 1 901€ | 2 236€ | 907€ | 12 272€ |
| Medical appointments | 6 764€ | 1 853€ | 2 140€ | 891€ | 11 648€ |
| Non-medical appointments | 464€ | 48€ | 96€ | 16€ | 624€ |
| Inpatient care | 46 524€ | 98 737€ | 18 033€ | 6 852€ | 170 146€ |
| Hematology | 46 524€ | 69 605€ | 18 033€ | 6 852€ | 141 014€ |
| ICU | NA | 29 132€ | NA | NA | 29 132€ |
| Diagnostic and therapeutic procedures | 51 735€ | 20 383€ | 23 369€ | 8 333€ | 103 820€ |
| Imaging exams (CT scan, PET-CT, MRI, bone scintigraphy) | 4 913€ | 5 710€ | 12 614€ | 3 098€ | 26 337€ |
| Cell therapy procedures | 26 374€ | 801€ | 47€ | NA | 27 222€ |
| Radiotherapy sessions | 2 760€ | NA | 1 756€ | 502€ | 5 018€ |
| Others (blood tests and exams) | 17 687€ | 13 872€ | 8 951€ | 4 733€ | 45 243€ |
| Sub-total cost | 105 486€ | 121 022€ | 43 638€ | 16 092€ | 286 238€ |
| Total cost (including CAR T-cell drug cost) | 6 995 444€ | 121 022€ | 43 638€ | 16 092€ | 7 176 196€ |

NA, not applicable; ICU, intensive care unit; CT, computed tomography; PET-CT, Positron Emission Tomography—Computed Tomography; MRI, Magnetic Resonance Imaging.

**Table 6. Costing by activity.**

| Activity | Median cost | Minimum cost | Maximum cost |
|---|---|---|---|
| Outpatient care (medical and non-medical appointments) | 536€ | 333€ | 1 179€ |
| Inpatient care | 6 131€ | 1 443€ | 29 945€ |
| Diagnostic and therapeutic procedures | | | |
| Imaging exams (CT scan, PET-CT, MRI, bone scintigraphy) | 1 033€ | 0€ | 3 098€ |
| Cell therapy procedures | 1 053€ | 556€ | 2 407€ |
| Radiotherapy sessions | 251€ | 0€ | 1 004€ |
| Others (blood tests and exams) | 1 663€ | 740€ | 8 457€ |
| Sub-total cost | 10 667€ | 3 072€ | 46 090€ |
| Total cost (including CAR T-cell drug cost) | 355 165€ | 347 570€ | 390 588€ |

Abbreviations: NA, not applicable; CT, computed tomography; PET-CT, Positron Emission Tomography—Computed Tomography; MRI, Magnetic Resonance Imaging.

between 2017 and 2020, while the length of stay was higher (ICU length of stay of 4 days in the aforementioned study) [38].

As observed in other cohorts, 10% of patients selected to this therapy progressed or deceased before harvesting cells by leukapheresis [3–5, 36]. Also, median hospital length of stay of 17 days for the procedure is in line with estimates described in others real world data cohorts [4, 5, 37, 39].

The higher number of patients selected for treatment with axi-cel in this cohort (65% vs. 35% with tisa-cel) was related to higher axi-cel availability to Early Access Program, in which pharmaceutical companies provide innovative drugs to patients before reimbursement issues were completed by health systems. It is worth highlighting that observed deaths mostly resulted from progression of the disease then from toxicity following CAR T cell infusion.

The therapeutic value of axi-cel and tisa-cel, accessed through cost-effectiveness analyses of health outcomes gained, has been widely recognized by health systems of several countries [11, 17, 29]. In Spain, cost-effectiveness analysis showed a greater benefit of 10.10 life-years gained and 8.97 quality-adjusted life-years (QALYs) gained of tisa-cel compared to salvage chemotherapy, associated to an additional 258 378€ total cost of tisa-cel, in relapsed/refractory B-cell acute lymphoblastic leukemia [40]. In the US, in patients with B-cell lymphoma, cost-effectiveness estimates of axi-cel ranged from $58 000 to $289 000/QALYs gained and estimates of tisa-cel ranged from $168 000 to $223 000/QALYs gained, depending on model assumptions relating to cost and long-term survival [17, 29, 41]. In a recent comprehensive systematic review of gene therapies cost effectiveness, all CAR T-cells therapies reviewed shown to be more effective than their comparators. However, not all CAR T-cell therapeutic regimens were cost effective at standard cost-effective thresholds. Specifically, these therapies shown to be more cost effective in conditions with significant risk of mortality and/or disability [20]. The very high cost of CAR T-cells has raised concerns for patient access to these therapies, and the financial sustainability of healthcare systems, and it has been argued that substantial price reductions, or price payment only for initial complete response (at current prices), is required to improve axi-cel and tisa-cel cost effectiveness [41].

We observed that list price of CAR T-cell therapy in IPO-Porto is within the range of prices reviewed for 6 European members states, ranging between 307 200€ and 350 000€ [11]. As reported in other countries, drug acquisition is the largest component of the cost of CAR T-cell therapy in IPO-Porto (97% of total cost for each patient treated) [11, 17, 39]. In our context, when drug acquisition cost was excluded, median total cost was 10 667€ per treated patient. In comparison to higher figures in other European countries, our lower cost estimate

can be explained by country specific prices but mostly by different methodological approaches as, for example, we did not consider costs attributable to adverse effects [11, 39]. In accordance with other studies, inpatient care had substantial impact on healthcare expenditure (nearly 58% of the total cost excluding CAR T-cell acquisition cost) followed by costs related to pathology and laboratory work and cell therapy procedures [11, 14, 18, 39].

In this cohort of patients with relapsed/refractory large B-cell lymphomas with indication for CAR T-cell therapy, the total cost of treatment was 7 176 196€, or 286 238€ when we excluded CAR T-cell production cost. In the future, the price for CAR T-cell therapies is expected to change due to additional new players adding competition to same disease indication [11]. The cost of care associated with this therapy may as well decrease over time due to reduction of adverse effects and growing management skills. Also, the potential availability of CAR T-cell therapies suitable for outpatient administration and a streamlined manufacturing process, may reduce total costs of care and make these therapies more cost-effective [14, 17].

The price for this therapy is mostly subject to confidential negotiations but several European Union countries, including Italy, Spain, and Portugal, adopted the outcome-based strategy to reimbursement negotiations [11]. In Portugal, the National Health Authority agreed in shared risk reimbursement model, paying subsets of each treatment according to OS for a specific patient, meaning that full payment will be delivered for more than 12 months follow-up for a particular patient/treatment. We hope to evaluate this model with our cohort in the future.

Limitations of this study are related its observational and retrospective design, lacking a prospective evaluation of costs, which could have resulted in underestimation of the real cost. Estimate the costs based on healthcare resources utilization may also have undervalued estimates because expenses resulting from treatments outside IPO-Porto and from procedures without regulated prices were not considered. Additionally, health resources unit prices from the year 2019 may not reflect the actual cost. The short follow-up period of this study precluded late adverse effects frequency and associated costs assessment, as well as how improvement in adverse effects management may impact in costs reduction.

In conclusion, our findings highlight the heavy economic burden related to CAR T-cell therapy largely attributable to drug acquisition costs.

Further prospective data will consolidate these results, adding quality of life evaluation and Markov model analysis. The shared risk reimbursement strategy adopted by National Health System will be further evaluated, helping to facilitate future decisions for health stakeholders, when adopting this technology.

## Supporting information

**S1 Fig. Patients flow diagram.** CAR, chimeric antigen receptor; DP, disease progression; ICANS, immune effector cell-associated neurotoxicity syndrome; CR, complete response; PR, partial response; HLH, Hemophagocytic lymphohistiocytosis.
(PDF)

## Author Contributions

**Conceptualization:** Sérgio Chacim, Maria José Bento, José Mário Mariz.

**Data curation:** Sérgio Chacim, José Luís Cunha, Pedro Medeiros.

**Formal analysis:** Teresa Monjardino, José Luís Cunha, Pedro Medeiros, Patrícia Redondo.

**Investigation:** Sérgio Chacim, Teresa Monjardino, Pedro Medeiros, Patrícia Redondo.

**Methodology:** Sérgio Chacim, Teresa Monjardino, Pedro Medeiros, Patrícia Redondo, Maria
José Bento, José Mário Mariz.

**Supervision:** Patrícia Redondo, Maria José Bento, José Mário Mariz.

**Validation:** Sérgio Chacim, Teresa Monjardino, José Luís Cunha, Pedro Medeiros, Patrícia
Redondo, Maria José Bento, José Mário Mariz.

**Visualization:** Sérgio Chacim, Teresa Monjardino, José Luís Cunha, Pedro Medeiros, Patrícia
Redondo, Maria José Bento, José Mário Mariz.

**Writing – original draft:** Sérgio Chacim, Teresa Monjardino.

**Writing – review & editing:** Sérgio Chacim, Teresa Monjardino, José Luís Cunha, Pedro
Medeiros, Patrícia Redondo, Maria José Bento, José Mário Mariz.

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
