## [Decision Letter · Decision Letter 0]

20 Sep 2022

PONE-D-22-18355Costs, effectiveness, and safety associated with Chimeric Antigen Receptor (CAR) T-cell therapy: results from a Comprehensive Cancer CenterPLOS ONE

Dear Dr. Monjardino,

Thank you for submitting your manuscript to PLOS ONE. After careful consideration, we feel that it has merit but does not fully meet PLOS ONE’s publication criteria as it currently stands. Therefore, we invite you to submit a revised version of the manuscript that addresses the points raised during the review process.

We look forward to receiving your revised manuscript.

Kind regards,

Yasunori Sato

Academic Editor

PLOS ONE

Journal Requirements:

Reviewers' comments:

Reviewer's Responses to Questions

**Comments to the Author**

1. Is the manuscript technically sound, and do the data support the conclusions?

Reviewer #1: Yes

Reviewer #2: Yes

Reviewer #3: Yes

2. Has the statistical analysis been performed appropriately and rigorously? 

Reviewer #1: No

Reviewer #2: Yes

Reviewer #3: Yes

3. Have the authors made all data underlying the findings in their manuscript fully available?

Reviewer #1: Yes

Reviewer #2: Yes

Reviewer #3: Yes

4. Is the manuscript presented in an intelligible fashion and written in standard English?

Reviewer #1: Yes

Reviewer #2: Yes

Reviewer #3: Yes

5. Review Comments to the Author

Reviewer #1: The articles in very valuable for those working in the field of hematology and for centres considering the implementation of CAR-T cell therapy.

Most centres and health care facilities consider the CAR-T therapy acquisition cost as the only determinant factor. The current data illustrates the auxiliary costs associated with diagnostics, hospitalization and management of complications and highlights the actual cost as well as the results in real-world and not in the context of pre-planned clinical trials.

The article also compares the cost to the cost of the procedure in different European countries.

The stratification of cost per stage as displayed in table 5 and by activity in table 6 is also of great importance and is very illustrative

very few corrections need to be made namely, a few abbreviations need to be highlighted PMBL, lin2 162, NT, line 182..

Reviewer #2: This is a very well written manuscript on the cost-effectiveness of commercial CART-cells. The authors must be commended for their detailed analysis of their own practice and how nicely the manuscript is structured.

I only have a few minor comments:

1. Response rate: the authors may choose whether the denominator is 20 (all patients included) or 18 (all patients treated), but never 16 (patients evaluated). Imagine now that I have a trial and 10 patients are treated: 8 patients are never evaluated for response because they die early or experience disease progression and 2 patients are evaluated. Both patients respond to therapy. You would never say that the response rate is 100%, right?

2. The sentence starting on line 188: "Four hospitalizations..." is unclear.

3. Line 227: "progressed/diseased" or "progressed/deceased"?

4. I would prefer the term "ICANS" to "NT".

5. Please define high-dose and low-dose corticosteroids. By the way, is this distinction important?

Reviewer #3: The manuscript describes costs of CAR-T cell therapy in DLBCL third line at a Portuguese CCC centre. The medical costs were as high as 97%.

The manuscript does not address the cost -effectiveness of the treatment as there is no historical comparison to a standard third line therapy before CAR-T, and the novelty of the manuscript is moderate. It is, however well performed. I am not an expert on cost estimations, but the methods seem reasonable. The efficacy of the treatment is hard to evaluate with a very short median follow up of 4 months.

Minor comments:

1.As this is an intention-to-treat analysis, I would prefer to add the 4 patients excluded (two died due to disease progression before CAR-T, two died after CAR T due to toxicity or progression.

2. Cellof origin is given for twelve patients with 8 missing. I think it is better to have two missing as COO seldom is given for PMBCL or transformed lymhomas (3 + 3 patients).

3. Bridging therapy is given, but I could not find lymphodepletion (I asssume FC or Bendamustin or neither)

4. Minor language deficiencies can be fixed during editorial review.

6. PLOS authors have the option to publish the peer review history of their article (what does this mean?). If published, this will include your full peer review and any attached files.

Reviewer #1: **Yes: **Hadi A. Goubran Messiha

Reviewer #2: **Yes: **Julio Delgado

Reviewer #3: No

---

## [Author Response · Author response to Decision Letter 0]

18 Nov 2022

RESPONSES TO THE COMMENTS FROM THE EDITORS AND REVIEWERS

We acknowledge the important issues that were raised by Editors and Reviewers, and we believe that the changes made to the manuscript resulted in substantial improvement of the overall quality and clarity of the work. Comments are individually addressed below.

Journal Requirements

R: We have changed the manuscript layout and files naming to meet the PLOS ONE style requirements. 

R: In the revised cover letter we transcribed the justification written by the Data Protection Officer from the Portuguese Oncology Institute of Porto (IPO-Porto) of the ethical and legal restrictions on sharing a de-identified data set. IPO-Porto Data Protection Officer provided the following explanation: “This study is based on a very small sample (n=20), so we understand that the particularity of this study, given the current state of the art, may lead to an increased risk of violating the privacy of our patients through a possible identification of the data subject. However, this concern is the result of the ethical and legal obligations to which our institution is bound in light of the current guidelines and recommendations and legislation”. We also updated the manuscript with institutional contacts information to which data requests may be sent as follows: Data requests may be sent to the Data Protection Officer (epd@ipoporto.min-saude.pt) and to the Department of Onco-hematology at IPO-Porto (s.oncohematologia@ipoporto.min-saude.pt).

R: We updated the reference list with three relevant recent publications and replaced the reference from an abstract presented in a conference with the recently published manuscript (Kuhnl et al 2022) as follows:

Gajra A, Zalenski A, Sannareddy A, Jeune-Smith Y, Kapinos K, Kansagra A. Barriers to Chimeric Antigen Receptor T-Cell (CAR-T) Therapies in Clinical Practice. Pharmaceut Med. 2022;36(3):163-71. Epub 20220607. doi: 10.1007/s40290-022-00428-w. PubMed PMID: 35672571; PubMed Central PMCID: PMC9217916.

Maziarz RT, Yang H, Liu Q, Wang T, Zhao J, Lim S, et al. Real-world healthcare resource utilization and costs associated with tisagenlecleucel and axicabtagene ciloleucel among patients with diffuse large B-cell lymphoma: an analysis of hospital data in the United States. Leuk Lymphoma. 2022;63(9):2052-62. Epub 20220414. doi: 10.1080/10428194.2022.2060503. PubMed PMID: 35422192.

Valade S, Darmon M, Zafrani L, Mariotte E, Lemiale V, Bredin S, et al. The use of ICU resources in CAR-T cell recipients: a hospital-wide study. Ann Intensive Care. 2022;12(1):75. Epub 20220817. doi: 10.1186/s13613-022-01036-2. PubMed PMID: 35976532; PubMed Central PMCID: PMC9385897.

Kuhnl A, Roddie C, Kirkwood AA, Tholouli E, Menne T, Patel A, et al. A national service for delivering CD19 CAR-Tin large B-cell lymphoma - The UK real-world experience. Br J Haematol. 2022;198(3):492-502. Epub 20220429. doi: 10.1111/bjh.18209. PubMed PMID: 35485402.

Reviewers' comments

Reviewer #1: 

We thank the reviewer for his careful reading of the manuscript and their constructive remarques.

[…] very few corrections need to be made namely, a few abbreviations need to be highlighted PMBL, lin2 162, NT, line 182.

R: Errors in abbreviations have been corrected accordingly.

Reviewer #2: 

1. Response rate: the authors may choose whether the denominator is 20 (all patients included) or 18 (all patients treated), but never 16 (patients evaluated). Imagine now that I have a trial and 10 patients are treated: 8 patients are never evaluated for response because they die early or experience disease progression and 2 patients are evaluated. Both patients respond to therapy. You would never say that the response rate is 100%, right?

R: We thank the reviewer for pointing this out. We have now followed the appropriate suggestion from two of the reviewers and updated the effectiveness analysis to include the 20 participants in the denominator, consistent with the definition an intention-to-treat analysis. 

2. The sentence starting on line 188: "Four hospitalizations..." is unclear.

R: We have rewritten the sentence in a different way to make it clearer.

3. Line 227: "progressed/diseased" or "progressed/deceased"?

R: The word diseased was corrected by deceased.

4. I would prefer the term "ICANS" to "NT".

R: We agreed that that NT is not the accurate term to describe these events. We replaced it with ICANS throughout the manuscript.

5. Please define high-dose and low-dose corticosteroids. By the way, is this distinction important?

R: We agree with the reviewer's comment, and we have added information about the definition for high-dose and low-dose corticosteroid use to the manuscript. High dose is defined as 1000mg of intravenous methylprednisolone as a single day dose or 1mg/kg every 12 hours and low dose corticosteroids is defined as dexamethasone up to 10mg every 6 hours or equivalent dose. 

In our point of view this distinction is relevant because the high dose of methylprednisolone 1000 mg/day used for grade 4 CRS/ICANS treatment, that is according to the current recommendations (Santomasso, B D, et al. Management of Immune-Related Adverse Events in Patients Treated With Chimeric Antigen Receptor T-Cell Therapy: ASCO Guideline, Journal of Clinical Oncology 2021 39:35, 3978-3992), is substantially higher than the corticosteroid dose recommended for the other situations.

Reviewer #3: 

1. As this is an intention-to-treat analysis, I would prefer to add the 4 patients excluded (two died due to disease progression before CAR-T, two died after CAR T due to toxicity or progression.

R: We thank the reviewer for pointing this out. We have now followed the appropriate suggestion from two of the reviewers and updated the effectiveness analysis to include the 20 participants in the denominator, consistent with the definition an intention-to-treat analysis. 

2. Cell of origin is given for twelve patients with 8 missing. I think it is better to have two missing as COO seldom is given for PMBCL or transformed lymphomas (3 + 3 patients).

R: We agree with the reviewer in this remark. We have changed table 1 to present the distribution of cell of origin exclusively for DLBCL patients.

3. Bridging therapy is given, but I could not find lymphodepletion (I assume FC or Bendamustin or neither)

R: We agree that the description of lymphodepletion chemotherapy is of utmost relevance to characterize costs associated to CAR T-cell therapies and that information was included in table 1 notes as follows. After scheduling the infusion day (day 0) lymphodepletion chemotherapy is administered in the following way: 

For Axicabtagene ciloleucel:

- intravenous ondansetron (8 mg) every 8 hours, at days -5, -4 and -3

- intravenous saline solution (2000 ml) per day, at days -5, -4 and -3

- intravenous cyclophosphamide (500 mg/m2) infused over 60 minutes, at days -5, -4 and -3

- intravenous fludarabine (30 mg/m2, maximum dose of 50 mg) infused over 30 minutes, at days -5, -4 and -3

For Tisagenlecleucel:

- intravenous ondansetron (8 mg) every 8 hours, at days -5, -4 and -3

- intravenous saline solution (2000 ml) per day, at days -5, -4 and -3

- intravenous cyclophosphamide (250 mg/m2) infused over 60 minutes, at days -5, -4 and -3

- intravenous fludarabine (25 mg/m2, maximum dose of 50 mg) infused over 30 minutes, at days -5, -4 and -3

4. Minor language deficiencies can be fixed during editorial review.

R: Inaccuracies in language and spelling capture by the reviewers were corrected Additional comments and suggestions are welcome and appreciated.

---

## [Editor Report · Decision Letter 1]

25 Nov 2022

Costs, effectiveness, and safety associated with Chimeric Antigen Receptor (CAR) T-cell therapy: results from a Comprehensive Cancer Center

PONE-D-22-18355R1

Dear Dr. Monjardino,

We’re pleased to inform you that your manuscript has been judged scientifically suitable for publication and will be formally accepted for publication once it meets all outstanding technical requirements.

Kind regards,

Yasunori Sato

Academic Editor

PLOS ONE
---

## [Editor Report · Acceptance letter]

1 Dec 2022

PONE-D-22-18355R1 

Costs, effectiveness, and safety associated with Chimeric Antigen Receptor (CAR) T-cell therapy: results from a Comprehensive Cancer Center 

Dear Dr. Monjardino:

I'm pleased to inform you that your manuscript has been deemed suitable for publication in PLOS ONE. Congratulations! Your manuscript is now with our production department. 

Kind regards, 

on behalf of

Dr. Yasunori Sato 

Academic Editor

PLOS ONE